# Snail Track Lesion with Flat Keratometry in Anterior Segment Dysgenesis Caused by a Novel *FOXC1* Variant

**DOI:** 10.3390/jcm11175166

**Published:** 2022-08-31

**Authors:** Pavlina Skalicka, Jana Jedlickova, Ales Horinek, Marie Trkova, Alice E. Davidson, Stephen J. Tuft, Lubica Dudakova, Petra Liskova

**Affiliations:** 1Department of Paediatrics and Inherited Metabolic Disorders, First Faculty of Medicine, Charles University and General University Hospital in Prague, 128 08 Prague, Czech Republic; 2Department of Ophthalmology, First Faculty of Medicine, Charles University and General University Hospital in Prague, 128 08 Prague, Czech Republic; 3Institute of Biology and Medical Genetics, First Faculty of Medicine, Charles University, Prague and General University Hospital in Prague, 128 08 Prague, Czech Republic; 4GENNET, 170 00 Prague, Czech Republic; 5UCL Institute of Ophthalmology, London EC1V 9EL, UK; 6Moorfields Eye Hospital, London EC1V 2PD, UK

**Keywords:** *FOXC1*, corneal dystrophy, anterior segment dysgenesis, keratometry, corneal endothelium

## Abstract

We report the phenotype of a 15-year-old female patient with anterior segment dysgenesis (ASD) caused by a novel heterozygous loss-of-function *FOXC1* variant. The proband underwent an ophthalmic examination as well as a molecular genetic investigation comprising exome sequencing, a single nucleotide polymorphism array to access copy number and Sanger sequencing to exclude non-coding causal variants. There was bilateral mild iris hypoplasia with pupil deformation and iridocorneal adhesions. In addition to these features of ASD, the corneas were flat, with mean keratometry readings of 38.8 diopters in the right eye and 39.5 diopters in the left eye. There was a snail track lesion of the left cornea at the level of the Descemet membrane. The central corneal endothelial cell density was reduced bilaterally at 1964 and 1373 cells/mm^2^ in the right and left eyes, respectively. Molecular genetic analysis revealed that the proband was a carrier of a novel heterozygous frameshifting variant in *FOXC1*, c.605del p.(Pro202Argfs*113). Neither parent had this change, suggesting a de novo origin which was supported by paternity testing. We found no possibly pathogenic variants in the other genes associated with posterior corneal dystrophies or ASD. Further studies are warranted to verify whether there is a true association between snail track lesions, corneal flattening, and pathogenic variants in *FOXC1*.

## 1. Introduction

Anterior segment dysgenesis (ASD) comprises a group of rare developmental disorders characterized by structural abnormalities of the cornea (opacity, microcornea, vascularization, posterior embryotoxon), iris (corectopia, polycoria, hypoplasia, coloboma, anterior synechiae), lens (ectopia lentis, microphakia, spherophakia, cataract), and the trabecular meshwork [1]. However, primary corneal endothelial disease is not usually a feature. Most of the changes are static but trabeculodysgenesis can result in secondary glaucoma [1]. ASD exhibits genetic and phenotypic heterogeneity with mutations in more than 10 genes, the majority of which encode transcription factors [2]. The most commonly occurring pathogenic variants are in *FOXE3*, *PAX6*, *PITX2,* and *FOXC1* [1,2].

Dominantly inherited mutations in *FOXC1* are associated with ASD type 3 (OMIM #601631) and Axenfeld-Rieger syndrome type 3 (OMIM #602482), characterized by additional systemic features of dental abnormalities, craniofacial dysmorphism, and hearing loss [3,4]. *FOXC1* encodes Forkhead Box C1, which belongs to the human Forkhead-box (FOX) family, many of which play a vital role in neural crest differentiation [5]. The expression of both *FOXC1* and *PITX2* is essential for the development of the corneal endothelium [6]. Their absence in the cells of the periocular mesenchyme results in a lack of either corneal endothelial growth or the formation of the anterior chamber [6,7].

There are several reports on patients suffering from both ASD and Fuchs endothelial corneal dystrophy (FECD), a condition that manifest as corneal guttae with a gradual loss of corneal endothelial cells [8,9,10]. The underlying genetic cause of FECD in approximately 75% of cases is an expansion of a triplet repeat situated within an intron of *TCF4* (termed CTG18.1) [11]. Variants in several other genes have also been identified, including *COL8A2*, *SLC4A11*, and *ZEB1*; however, several of these findings await replication [12].

Linear snail (or rail) track lesions are characteristic posterior corneal structures with scalloped edges at the level of the Descemet membrane and endothelium. They have been reported as a feature of posterior corneal vesicles (PCVs) [13]. They show phenotypic overlap with posterior polymorphous corneal dystrophy (PPCD) caused by pathogenic variants in *OVOL2*, *ZEB1,* and *GRHL2* [14,15,16,17], but no genetic cause for PCVs has been identified [13], although an association with x-linked megalocornea was noted in one case [18]. Corneas that are flatter than normal are characteristic of cornea plana, which is associated with mutations in *KERA* [19].

In this report, we describe a case of ASD caused by a *FOXC1* variant in which there was a marked keratometric flattening of the cornea associated with a unilateral snail track lesion and a decrease in corneal endothelial cell density.

## 2. Materials and Methods

The detailed ophthalmic examination performed in the proband included the measurement of the best-corrected Snellen visual acuity (BCVA) extrapolated to decimal values, non-contact ocular tonometry, corneal tomography (Pentacam, Oculus, Wetzlar, Germany) to provide information on corneal thickness and keratometry, non-contact specular microscopy (Noncon ROBO Pachy SP-9000, Konan Medical Inc., Tokyo, Japan) to measure the corneal endothelial cell density, and spectral-domain optical coherence tomography (SD-OCT; Spectralis, Heidelberg Engineering GmbH, Heidelberg, Germany) to assess the cornea, macula, and peripapillary retinal nerve fiber layer (RNFL) thickness (circular scan centered on the optic disc within a 3.5–3.6 mm diameter). The axial length of the eyes was measured with an IOL-Master V.5 (Carl Zeiss Meditec AG, Jena, Germany). We used a Medmont M700 (Medmont Pty Ltd., Nunawading, Australia) instrument to examine the visual fields. We also performed standard audiometry to assess hearing.

Genomic DNA was extracted from venous blood (Gentra Puregene Blood Kit, Qiagen, Hilden, Germany) according to the manufacturer’s instructions. We performed exome sequencing using a SureSelect Human All Exome V6 capture kit (Agilent, Santa Clara, CA, USA). Generated libraries were sequenced on NovaSeq 6000 (Illumina, San Diego, CA, USA). Raw sequencing data were annotated using Franklin by Genoox (https://franklin.genoox.com, accessed on 12 July 2022) which also provides automatic workflow of FASTQ data to a shortlist of candidate variants [20]. The average variant depth was 77 and the percentage of targeted positions covered by at least 20 reads was 83.6%. Rare variants with minor allele frequencies, ≤0.005 as per gnomAD v2.1.1 [21], in genes known to be associated with structural eye disease (PanelApp, version 1.132) and corneal dystrophy genes (version 1.6) were filtered [22]. Next, we used Sanger sequencing to screen the *OVOL2* and *GRHL2* regulatory regions (not covered by exome sequencing) previously implicated in the pathogenesis of PPCD types 1 and 4, respectively [14,15]. In addition, CTG18.1 was genotyped using the short tandem repeat assay [11].

A copy number variations (CNVs) assessment was performed using an Infinium Global Screening Array–24 v2.0 (650K, Illumina, San Diego, USA). Results were analysed by a cnvPartition 3.2.0 algorithm implemented in Illumina Genome Studio v2.0 software. Identified CNVs were mapped against human genome assembly hg19. Paternity testing was performed with a set of 16 forensic markers [23].

## 3. Results

The 15-year-old female proband had no family history of ocular disease. Six months previously, she had an attack of left anterior uveitis with mildly elevated intraocular pressure (IOP). The BCVA was 1.0 in the right eye with a refraction of (−0.75/−0.75 × 28°) and 1.0 (plano refraction) in the left eye. In both eyes, the corneal shape was abnormal. There was also corectopia with mild underdevelopment of the iris stroma and iris strands touching the corneal endothelium in the inferior nasal quadrants (Figure 1B,C). The horizontal corneal diameters were 12.2 mm and 12.4 mm in the right and left eyes, respectively. In addition, in the left eye, there was a prominent snail track lesion (Figure 1D). The central endothelial cell densities were 1964 cells/mm^2^ in the right eye and 1373 cells/mm^2^ in the left eye, while adjacent and inferiorly to the snail track lesion, the cell density was 1122 cells/mm^2^ (Figure 1E,F) (the normal endothelial cell density values in individuals under 25 years of age are 2940 (±345) cells/mm^2^) [24].

Corneal tomography showed abnormally low keratometry values of K1 (flat meridian) 37.9 D and K2 (steep meridian) 39.7 D in the right eye, and K1 38.8 D and K2 40.2 D in the left eye, while central corneal thickness was within the normal range 551 μm and 566 μm, in the right and left eye, respectively (normal values between the 2.5 and 97.5 percentiles: 40.7–46.1 D for K1, 41.76–47.4 D for K2, ≥479 μm and ≤602 μm) [25]. The axial lengths were normal at 24.43 mm and 24.48 mm in the right and left eye, respectively (the normal range in younger adults is 23.60 ± 1.15 mm) [26]. The IOP was 21 mmHg in the right eye and 19 mmHg in the left eye and the visual fields were normal. There was no pathologic optic disc cupping. However, because the RNFL thickness was borderline (Figure 1I), the patient was advised to attend a follow-up visit for repeat IOP and RNFL measurements and visual field testing.

The patient was otherwise well with normal developmental milestones, no dental anomalies, and normal bilateral hearing. There was mild craniofacial dysmorphism (hypertelorism, broad flat nasal bridge). Thus, based on the observed phenotype, her clinical diagnosis could be classified as Axenfeld-Rieger syndrome.

We identified a novel *FOXC1* heterozygous point mutation c.605del p.(Pro202Argfs*113) (reference sequence NM_001453.3) (Figure 1A). The variant is predicted to result in a frameshift of the *FOXC1* coding sequence that would disrupt the 553 amino acids-long protein inhibitory domain (aa 215–365).

The variant was absent in the parents and an unaffected sister. Paternity and maternity testing further supported its de novo origin. Based on the American College of Medical Genetics and Genomics (ACMG) guidelines, the variant was classified as pathogenic, meeting the following criteria: PVS1 (null variant in a gene where the loss of function is a known mechanism of disease), PM2 (allele frequency is extremely low in all population databases), PS2 (de novo in a patient with phenotype consistency, no family history, and both maternity and paternity are confirmed), and PP4 (the patient’s phenotype or family history is highly specific for a disease with a single genetic etiology) [27]. There were no rare variants in genes associated with corneal endothelial dystrophies or cornea plana. The CTG18.1 repeat lengths were identified to be in a non-disease-associated range, 12 and 23, on each respective allele [11].

## 4. Discussion

We describe a patient with features of ASD caused by a novel de novo *FOXC1* variant classified as pathogenic and predicted to cause loss of function. Given the frequency of whole *FOXC1* gene deletions, haploinsufficiency is a clear mechanism for a *FOXC1*-associated disease [4].

Interestingly, the patient also had bilateral corneal flattening, as well as a unilateral corneal snail track lesion. The common corneal changes of ASD are abnormalities of the corneal shape or microcornea, corneal opacity, and posterior embryotoxon [3,9,28]. Although the corneal endothelium is not reported to be affected in the majority of cases with Axenfeld-Rieger syndrome, most studies lack a detailed genotype–phenotype characterization of the keratometry and endothelial cell density. To exclude other corneal conditions with phenotypic overlap (PPCD and cornea plana), or which were reported in association with ASD (FECD), we have screened the known genetic causes [11,18,29]. No rare variants or deletions were detected [14,15,30,31], and CTG18.1 bi-allelic repeat lengths were both in the normal range [11]. However, we cannot exclude the possibility that there are alternative genetic causes of the corneal snail track lesion located either in the coding or non-coding part of the genome.

Given FOXC1 is essential for the differentiation of the mesenchymal cells of the anterior segment that are responsible for the development of the cornea, iris, and aqueous drainage structures [6,7], and because the current case has flat corneas with a bilaterally decreased endothelial cell density and a unilateral snail track, more typically found in corneas with PCVs, we hypothesize that these features may represent rare features associated with *FOXC1* mutations. However, further studies are required to confirm this association.

## Figures and Tables

**Figure 1 jcm-11-05166-f001:**
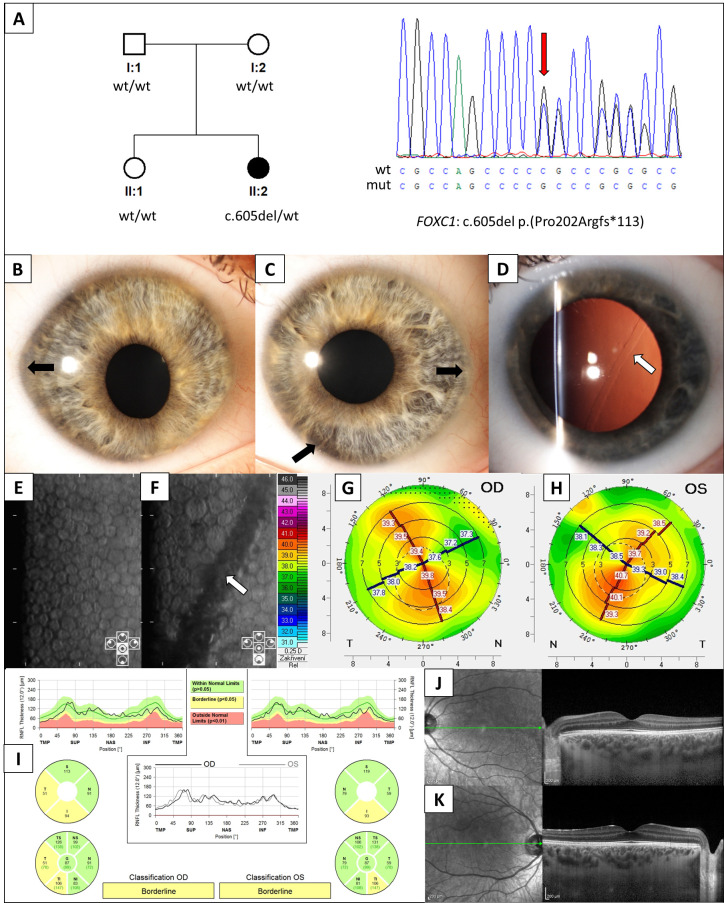
**Genetic testing and ocular findings in an individual with anterior segment dysgenesis.** (**A**) Pedigree of the family and sequence chromatogram highlighting *FOXC1* mutation detected in the proband (red arrow). The right (**B**) and left (**C**) anterior segment photographs taken under direct diffuse illumination. Note the abnormal elliptical corneal shape and pupil displacement (corectopia), particularly in the right eye. There is also mild peripheral iris hypoplasia (black arrows). (**D**) The left cornea has an oblique snail track lesion at the level of the Descemet membrane (white arrow) demonstrated by retroillumination after pupil dilation. Specular microscopic imaging of the right (**E**) and left (**F**) corneas demonstrates a lower endothelial cell density (1964 cell/mm^2^) in the right eye and in the left eye (1122 cells/mm^2^) next to the linear snail track lesion passing obliquely across the image (white arrow). Front sagittal curvature tomography map of the right (**G**) and left (**H**) corneas. There is regular oblique astigmatism (the axes are symmetric, but the principal meridians are not at 90˚ or 180˚) with relatively flat corneas (mean keratometry of 38.8 diopters in the right eye and 39.5 diopters in the left eye). (**I**) Peripapillary retinal nerve fiber layer (RNFL) thickness evaluation using spectral domain ocular coherence tomography (SD-OCT), with bilateral borderline thinning in the temporal and inferior areas when compared to the Spectralis normative database. SD-OCT scans of the (**J**) right and (**K**) left macula show a normal retinal structure.

## Data Availability

Not applicable.

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
