# Peer review of "Snail Track Lesion with Flat Keratometry in Anterior Segment Dysgenesis Caused by a Novel FOXC1 Variant"

_jcm, 2022, doi:10.3390/jcm11175166_

Round 1

Reviewer 1 Report

This is an interesting paper reporting the phenotype of a 15-year-old female patient with anterior segment dysgenesis caused by a "de novo" heterozygous loss of function FOXC1 variant, associated with snail track lesion and flat keratometry.

This is a really challenging and unique case report. However, some changes are needed to improve the manuscript:

- In the Abstract, please specify what SNP stands for.

- Lines 36-41: please, add a reference for this sentence.

- Line 67: please, replace "a" with "an".

- In the Materials and Methods section (lines 74-75), the authors stated that they performed SD-OCT. However, in the Results section, no information was given on this exam (macula, optic nerve, RNFL, ecc.). The authors should provide this information to better describe their patient in this case report, also adding an image.

- In the Results section, the authors stated that the visual fields were normal (line 115). How they evaluated this? They should add this information in the Materials and Methods section.

Author Response

Reviewer 1

Comments and Suggestions for Authors

This is an interesting paper reporting the phenotype of a 15-year-old female patient with anterior segment dysgenesis caused by a "de novo" heterozygous loss of function FOXC1 variant, associated with snail track lesion and flat keratometry.

This is a really challenging and unique case report. However, some changes are needed to improve the manuscript:

  1. In the Abstract, please specify what SNP stands for.
  2. Lines 36-41: please, add a reference for this sentence.
  3. Line 67: please, replace "a" with "an".

RESPONSE: We would like to thank to the reviewer for spotting these errors, which have been corrected.

  1. In the Materials and Methods section (lines 74-75), the authors stated that they performed SD-OCT. However, in the Results section, no information was given on this exam (macula, optic nerve, RNFL, etc.). The authors should provide this information to better describe their patient in this case report, also adding an image.

RESPONSE: Using SD-OCT we have performed both anterior and posterior segment imaging, corneal scans were uninformative, however RNFL showed borderline thickness in temporal and inferotemporal region. To address this comment we have included the RNFL and macular scans into Figure 1.

  1. In the Results section, the authors stated that the visual fields were normal (line 115). How they evaluated this? They should add this information in the Materials and Methods section.

RESPONSE: We have added to the manuscript following text (lines 81–83):

“We used a Medmont M700 (Medmont Pty Ltd, Nunawading, Australia) instrument to examine the visual fields.”

Reviewer 2 Report

The authors Skalicka et al. report a novel FOXC1 frame-shift variant in a patient with anterior segment dysgenesis. The authors claim association of this FOXC1 frame-shift variant (c.605del p.(Pro202Argfs*113) to unilateral snail track lesion and flat corneal keratometry observed in anterior segment dysgenesis. Given that most corneal dystrophies (including anterior segment dysgenesis) are multifactorial in nature, I believe this study helps in unraveling the genes that may be contributing to the disease pathology. The authors performed clinical examination on the proband that involved measurement of visual acuity, corneal shape and thickness using corneal tomography, corneal endothelial cell count using specular microscopy, axial length of the cornea, and audiometry for hearing loss. The authors also measured an unknown parameter using spectral domain optical coherence tomography (SD-OCT). To establish a genetic correlation, genomic DNA was extracted from venous blood and genomic analysis was performed for variant identification. They used whole exome sequencing and Sanger sequencing on the OVOL2 and GRHL2 regulatory regions for disease-associated variant identification. Overall, this study is an important step in identifying causative genes associated with ASD. As a result, I would like to suggest a few improvements to the manuscript listed below: 

1.     I recommend describing the ophthalmic tests in methods (with some detail) for our readership not familiar with clinical ophthalmology. For example, the parameters the corneal tomography, specular microscopy, or SD-OCT measure.

2.     For the genomic DNA analysis, please include more details in the methods on how the quality checks on reads and mapping were performed. 

3.     When you say that the genes associated with structural eye diseases and corneal dystrophies were filtered, do you mean that these were the only loci examined? If yes, can there be an SNP in the non-coding regions that may be relevant to the patient phenotype? Please comment in discussion.

4.     Please explain the method used for IOP measurement.

5.     I recommend including a table for highlighting results of the quantitative ophthalmic exams (visual acuity, specular counts, corneal thickness, corneal diameter, axial length, intraocular pressure etc.) to make it easier for readers. 

6.     Please explain the significance of this FOXC1 frame-shift variant on the protein structure. Does it lead to a truncated protein? In this regard, I recommend incorporating amino acid change in Fig 1A along with nucleotide changes. Usually, studies show a stick figure comparing the normal protein to the variant.

7.     Given the IOP was higher than normal, can the authors please comment on whether they found any glaucoma-related phenotypes.  

8.     Please explain the SD-OCT analysis and the parameters tested. Did you find retinal or optic nerve head abnormalities given the high IOP? If available, it would be great to see retinal OCT scans.

9.     (Line 101) Fig. 1A,B labeled incorrectly.

10.  Please rewrite the legend for Figure 1 to include more information on each sub-figure.

11.  Please add markers on Fig. 1B,C to convey meaningful information like corectopia, iris hypoplasia, stand touching etc. It is also recommended to explain these terms in detail in the results sections (Line 100-102).

12.  (Line 103) Fig. 1C labeled incorrectly for snail track lesion.

13.  Please mark Fig. 1G,H (or explain in figure legend) to show the regular oblique astigmatism.

14.  Please explain the variant classification criteria and abbreviations PVS1, PM2, PM6, and PP5.

15.  (Line 125) Grammatical error 

16.  Given that the genomic analysis is the central part of the study and it claims association of FOXC1 frame-shift variant to disease pathology, I recommend providing more information of the analysis in the methods and results.

17.  In discussion, please expand on the developmental role of FOXC1 in the cornea and how the disease variant may be contributing to the snail track lesion. 

Author Response

Reviewer 2

Comments and Suggestions for Authors

The authors Skalicka et al. report a novel FOXC1 frame-shift variant in a patient with anterior segment dysgenesis. The authors claim association of this FOXC1 frame-shift variant (c.605del p.(Pro202Argfs*113) to unilateral snail track lesion and flat corneal keratometry observed in anterior segment dysgenesis. Given that most corneal dystrophies (including anterior segment dysgenesis) are multifactorial in nature, I believe this study helps in unravelling the genes that may be contributing to the disease pathology. The authors performed clinical examination on the proband that involved measurement of visual acuity, corneal shape and thickness using corneal tomography, corneal endothelial cell count using specular microscopy, axial length of the cornea, and audiometry for hearing loss. The authors also measured an unknown parameter using spectral domain optical coherence tomography (SD-OCT). To establish a genetic correlation, genomic DNA was extracted from venous blood and genomic analysis was performed for variant identification. They used whole exome sequencing and Sanger sequencing on the OVOL2 and GRHL2 regulatory regions for disease-associated variant identification. Overall, this study is an important step in identifying causative genes associated with ASD. As a result, I would like to suggest a few improvements to the manuscript listed below: 

  1. I recommend describing the ophthalmic tests in methods (with some detail) for our readership not familiar with clinical ophthalmology. For example, the parameters the corneal tomography, specular microscopy, or SD-OCT measure.

RESPONSE: To address this comment we have amended the text in the Method section as follows (lines 72-83):

“Detailed ophthalmic examination performed in the proband included measurement of the best-corrected Snellen visual acuity (BCVA) extrapolated to decimal values, non-contact ocular tonometry, corneal tomography (Pentacam, Oculus, Wetzlar, Germany) to provide information on corneal thickness and keratometry, non-contact specular microscopy (Noncon ROBO Pachy SP-9000, Konan Medical Inc., Tokyo, Japan) to measure the corneal endothelial cell density, and spectral-domain optical coherence tomography (SD-OCT; Spectralis, Heidelberg Engineering GmbH, Heidelberg, Germany) to assess the cornea, macula and peripapillary retinal nerve fiber layer (RNFL) thickness (circular scan centered on the optic disc within a 3.5-3.6 mm diameter). The axial length of the eyes was measured with an IOL-Master V.5 (Carl Zeiss Meditec AG, Jena, Germany). We used a Medmont M700 (Medmont Pty Ltd, Nunawading, Australia) instrument to examine the visual fields.”

  1. or the genomic DNA analysis, please include more details in the methods on how the quality checks on reads and mapping were performed. 

RESPONSE: The text has been modified as follows to incorporate this suggestion (lines 88-91):

“Raw sequencing data were annotated using Franklin by Genoox (https://franklin.genoox.com), which also provides automatic workflow of FASTQ data to a shortlist of candidate variants [20]. Average variant depth was 77 and percentage of targeted positions covered by at least 20 reads was 83.6%.”

  1. When you say that the genes associated with structural eye diseases and corneal dystrophies were filtered, do you mean that these were the only loci examined? If yes, can there be an SNP in the non-coding regions that may be relevant to the patient phenotype? Please comment in discussion.

RESPONSE: To address this comment we have added to the discussion section the following text (lines 179-181):

“However, we cannot exclude the possibility that there are alternative genetic causes of the corneal snail track lesion located either in coding or non-coding part of the genome.”

  1. Please explain the method used for IOP measurement.

RESPONSE: To address this comment we have added to the Methods section (lines 72-74):

“Detailed ophthalmic examination performed in the proband included measurement of the best-corrected Snellen visual acuity (BCVA) extrapolated to decimal values, non-contact ocular tonometry…”

  1. I recommend including a table for highlighting results of the quantitative ophthalmic exams (visual acuity, specular counts, corneal thickness, corneal diameter, axial length, intraocular pressure etc.) to make it easier for readers. 

RESPONSE: Thank you for this suggestion. However, as the values are all incorporated in the main text, we consider that this would be an unnecessary repetition.

  1. Please explain the significance of this FOXC1 frame-shift variant on the protein structure. Does it lead to a truncated protein? In this regard, I recommend incorporating amino acid change in Fig 1A along with nucleotide changes. Usually, studies show a stick figure comparing the normal protein to the variant.

RESPONSE: Position of the first amino acid changed is shown as a part of standard description in Figure 1. As we have recently published figure of the protein structure in relation to other detected pathogenic variants in FOXC1, and the paper is cited within the text (reference #4), we would prefer to reference this information rather than include it again in the text.

To address this comment we have added to the Result section (lines 133-135):

“The variant is predicted to result in a frameshift of the FOXC1 coding sequence that would disrupt the 553 amino acids long protein before the inhibitory domain (aa 215–365).”

And to the Discussion section (lines 166-169):

“We describe a patient with features of ASD caused by a novel de novo FOXC1 variant classified as pathogenic, predicted to cause loss-of-function. Given the frequency of whole FOXC1 gene deletions, haploinsufficiency is a clear mechanism for FOXC1-associated disease [4].”

  1. Given the IOP was higher than normal, can the authors please comment on whether they found any glaucoma-related phenotypes.  

RESPONSE: We have added to the Results section following text (lines 123-127):

“The IOP was 21 mmHg in the right eye and 19 mmHg in the left eye and the visual fields were normal. There was no pathologic optic disc cupping. However, because the RNFL thickness was borderline (Fig. 1I) the patient was advised to attend a follow-up visit for repeat IOP and RNFL measurements and visual field testing.“

We have also added RNFL and macular scans to Figure 1.

  1. Please explain the SD-OCT analysis and the parameters tested. Did you find retinal or optic nerve head abnormalities given the high IOP? If available, it would be great to see retinal OCT scans.

RESPONSE: We did not find optic nerve head abnormalities. We have added RNFL and macular scans to Figure 1.

  1. (Line 101) Fig. 1A,B labelled incorrectly.

RESPONSE: This error has been corrected.

  1. Please rewrite the legend for Figure 1 to include more information on each sub-figure.

RESPONSE: The legend of Figure 1 was rewritten as follows:

„Figure 1. Genetic testing and ocular findings in an individual with anterior segment dysgenesis. (A) Pedigree of the family and sequence chromatogram highlighting FOXC1 mutation detected in the proband. The right (B) and (C) left anterior segment photograph taken under direct diffuse illumination. Note the abnormal elliptical corneal shape and pupil displacement (corectopia), in particular in the right eye. There is also mild peripheral iris hypoplasia (black arrows). (D) The left cornea has an oblique snail track lesion at the level of Descemet membrane (white arrow) demonstrated by retroillumination after pupil dilation. Specular microscopic imaging of the right (E) and left (F) corneas demonstrates a lower endothelial cell density (1,964 cell/mm2) in the right eye, and in the left eye (1,122 cells/mm2) next to the linear snail track lesion passing obliquely across the image (white arrow). Front sagittal curvature tomography map of the right (G) and left (H) corneas. There is a regular oblique astigmatism (the axes are symmetric but the principal meridians are not at 90Ëš or 180Ëš) with relatively flat corneas (mean keratometry 38.8 diopters in the right eye and 39.5 diopters in the left eye). (I) Peripapillary retinal nerve fiber layer (RNFL) thickness evaluation using spectral domain ocular coherence tomography (SD-OCT), with bilateral borderline thinning in the temporal and inferior areas when compared to the Spectralis normative database. SD-OCT scans of the (J) right and (K) left macula show normal retinal structure.“

  1. Please add markers on Fig. 1B,C to convey meaningful information like corectopia, iris hypoplasia, stand touching etc. It is also recommended to explain these terms in detail in the results sections (Line 100-102).

RESPONSE: We have added black arrows to mark hypoplasia to Figure 1. We have also changed the figure legend text to (line 152-153:

“Note the abnormal elliptical corneal shape and pupil displacement (corectopia), in particular in the right eye.”

  1. (Line 103) Fig. 1C labelled incorrectly for snail track lesion.

RESPONSE: Thank you for spotting the error which has been corrected

  1. Please mark Fig. 1G,H (or explain in figure legend) to show the regular oblique astigmatism.

RESPONSE: Following text has been added to the Figure legend (line 159-161):

“There is a regular oblique astigmatism (the axes are symmetric but the principal meridians are not at 90Ëš or 180Ëš) with relatively flat corneas (mean keratometry 38.8 diopters in the right eye and 39.5 diopters in the left eye).“

  1. Please explain the variant classification criteria and abbreviations PVS1, PM2, PM6, and PP5.

RESPONSE: We have reviewed this part of the text and made following changes in the Results section (lines 137-143):

“Based on the American College of Medical Genetics and Genomics (ACMG) guidelines, the variant was classified as pathogenic, meeting following criteria: PVS1 (null variant in a gene where loss of function is a known mechanism of disease), PM2 (allele frequency is extremely low in all population databases, PS2 (de novo in a patient with phenotype consistency, no family history and both maternity and paternity are confirmed), and PP4 (patient's phenotype or family history is highly specific for a disease with a single genetic etiology) [27].”

  1. (Line 125) Grammatical error 

RESPONSE: Thank you for spotting the error which has been corrected

  1. Given that the genomic analysis is the central part of the study and it claims association of FOXC1 frame-shift variant to disease pathology, I recommend providing more information of the analysis in the methods and results.

RESPONSE: More information has been provided, we also added reference to our previous work (lines 166-169):

“We describe a patient with features of ASD caused by a novel de novo FOXC1 variant classified as pathogenic, predicted to cause loss-of-function. Given the frequency of whole FOXC1 gene deletions, haploinsufficiency is a clear mechanism for FOXC1-associated disease [4].”

  1. In discussion, please expand on the developmental role of FOXC1 in the cornea and how the disease variant may be contributing to the snail track lesion. 

RESPONSE: We address this issue in the Discussion section (lines 183-189):

“Given FOXC1 is essential for the differentiation of the mesenchymal cells of the anterior segment that are responsible for the development of the cornea, iris, and aqueous drainage structures [6,7], and because the current case has flat corneas with bilaterally decreased endothelial cell density and a unilateral snail track, more typically found in corneas with PCVs, we hypothesize that these features may represent rare features associated with FOXC1 mutations. However, further studies are required to confirm this association.”